# OFS: OVERLAPPED FEATURE SYNTHESIS FOR COMMUNICATION-EFFICIENT FEDERATED LEARNING

## ABSTRACT

Federated learning (FL) is deemed as a promising privacy-preserving distributed learning paradigm for decentralized non-IID data. However, it inevitably introduces significant communication overhead caused by frequent gradient exchanges, limiting its scalability. To mitigate this issue, recent work proposes to use data distillation for communication costs reduction. Yet, the existing approaches fail to fully exploit the distilled features, resulting in a suboptimal compression ratio. In this paper, we propose a novel method—Overlapped Feature Synthesis(OFS)—that enables global feature sharing during compression, enhancing both communication efficiency and model performance. Specifically, we introduce a global feature Sampler, which extracts several small feature maps from a large global feature map to enable parameter sharing. To balance global and personalized parameters, an offset coefficient and multiple sampling strategies are introduced to allow for a flexible trade-off between compression efficiency and model performance.Extensive experiments demonstrate that OFS achieves better convergence with a lower compression rate compared to competing methods. Compared to state-of-the-art data distillation methods, our approach achieves an approximately 1% improvement in accuracy while maintaining a 10% higher compression rate. Moreover, we conduct ablation studies and visualizations to investigate the effects of the offset coefficient, the number of clients, and the number of local training epochs on the effectiveness of our method. Furthermore, we analyze the relationship between global and personalized model parameters.

## 1 INTRODUCTION

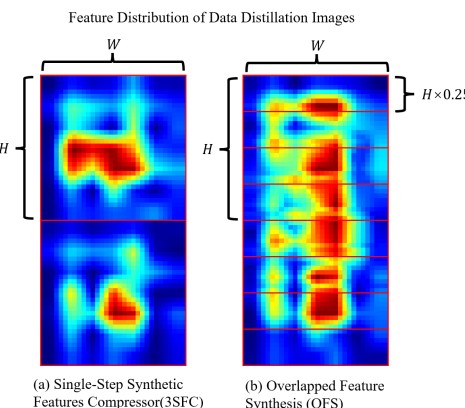

Figure 1: Feature distribution of data distillation images using (a) Single-Step Synthetic Features Compressor (3SFC) and (b) Overlapped Feature Synthesis (OFS)

Federated Learning (FL) McMahan et al. (2017) has emerged as a promising distributed learning paradigm to address data silos while preserving user privacy Kairouz et al. (2021). FL enables multiple clients to collaboratively train a global model without centralizing their local data, thereby mitigating privacy risks. However, frequent exchanges of gradient updates between the central server and clients introduce substantial communication overhead, particularly in large-scale models and datasets Li et al. (2020). As a result, reducing communication costs while maintaining model performance remains a key challenge in FLBonawitz et al. (2019).

To address this challenge, various compression methods such as sparsification Aji & Heafield (2017), quantization Bernstein et al. (2018); Alistarh et al. (2017), and data distillation Wang et al. (2020a) have been proposed. Among them, data distillation-based methods have gained increasing attention for their ability to synthesize compact representative data to replace raw gradients during transmission. These methods, such as Single-Step Synthetic Feature Compression (3SFC) Zhou et al. (2023), have demonstrated remarkable potential in achieving high compression rates without significantly compromising model accuracy.

However, a critical challenge across most data distillation methods lies in their limited ability to exploit inter-sample semantic redundancy Hooker et al. (2019). Despite their compactness, the synthetic datasets generated often contain repetitive or overlapping information that is not efficiently reused, leading to suboptimal communication efficiency and reduced performance gains. Our analysis using visualization tools such as Grad-CAM++ Selvaraju et al. (2017); Chattopadhay et al. (2018), taking 3SFC as an example, confirms the presence of redundant features across different synthetic samples, as illustrated in Figure 1,a.

To overcome this fundamental limitation, we propose a novel framework, Overlapped Feature Synthesis (OFS), which introduces an overlapping mechanism that structurally decomposes each synthetic sample into globally shared and locally personalized components. This strategy allows different samples to reuse shared semantics while retaining unique traits, significantly improving information utilization and reducing redundant transmission. Unlike traditional compression techniques or straightforward extensions of existing methods, OFS presents a conceptually distinct approach that enhances both compression efficiency and model performance. Moreover, the overlapping structure is non-trivial to integrate into standard compressors like sparsification or quantization, highlighting the independent methodological contribution of our work. As illustrated in Figure 1, OFS represents a generic and extensible strategy that addresses a core weakness in data distillation-based communication schemes.

We conduct extensive experiments to evaluate our method, demonstrating that the introduction of shared components effectively reduces communication overhead while improving model performance.

Our contributions are summarized as follows:

1. We observe that data distillation methods, exemplified by 3SFC, exhibits significant redundancy in compressed images and lacks support for continuous compression, limiting its scalability. To address this issue, we propose a novel approach that enables multiple images to share global parameters, reducing redundancy and improving compression efficiency.

2. We explore various overlap strategies, dividing compressed images into globally shared and personalized parameters. This approach refines the granularity of compression and enhances data transmission efficiency, further alleviating redundancy issues.

3. We validate our proposed method through extensive experiments on multiple datasets, models, and experimental settings. The results show substantial improvements in model accuracy even under constrained communication budgets. Additionally, we conduct ablation studies, visual analyses, and quantitative evaluations to further demonstrate the benefits of our method. The code will be open-sourced after publication.

## 2 RELATED WORK

**Sparsification.** Sparsification techniques have been widely used to reduce communication overhead in distributed learning by selectively transmitting only the most significant gradients. Typical methods include random-k and top-k gradient selection strategies. For instance, Strom Ström (2015) proposed threshold quantization that sends only gradients exceeding a predefined threshold, while Aji and Heafield Aji & Heafield (2017) introduced gradient dropping to discard insignificant gradients. Deep Gradient Compression (DGC)Lin et al. (2017) achieves extremely high compression ratios (up to 600×) without compromising model accuracy or convergence by compressing gradients globally. Top-k sparsification further improves communication efficiency by transmitting only the largest-magnitude gradients, providing strong theoretical convergence guaranteesWangni et al. (2018). Despite their effectiveness, sparsification methods face challenges in federated learning with many clients, since distinct sparse patterns from clients hinder efficient downlink compression and aggregation Reisizadeh et al. (2020).

**Quantization.** Quantization reduces communication costs by representing gradients with lower bit-width formats, such as 1-bit or 3-bit encodings. Notable approaches include 1-bit SGD Bernstein et al. (2018), which compresses gradients to binary values for faster transmission, and QSGD Alistarh et al. (2017), which trades off precision and communication efficiency. TernGrad Wen et al. (2017) introduced ternary quantization for convolutional networks, improving convergence under

certain scenarios. Recent advances like Z-sign Tang et al. (2024) mitigate gradient bias by adding symmetric noise perturbations and propose z-SignFedAvg, a sign-based federated averaging algorithm supporting multiple local updates to enhance communication efficiency. Error compensation methods Wu et al. (2018) also improve quantization performance by correcting for approximation errors. While quantization methods effectively reduce bandwidth, their compression ratios are generally lower than sparsification, limiting their gains in low-bandwidth or large-scale settings.

**Data Distillation.** Data distillation methods compress gradient information by synthesizing small representative datasets that approximate the original gradients Goetz & Tewari (2020); Hu et al. (2022); Zhu et al. (2021). These approaches optimize synthetic data to minimize the $\ell_2$ distance to the real model gradients. However, such methods can be computationally expensive and unstable when scaling to large models and datasets, occasionally causing system failures. The Single-Step Synthetic Feature Compression (3SFC) Zhou et al. (2023) addresses these issues by employing a similarity-based objective that requires only a single optimization pass, significantly enhancing both stability and efficiency, particularly under low compression ratios. Nonetheless, 3SFC still suffers from redundancy in synthesized features and limited compression efficiency, thereby motivating further refinement.

## 3 PROBLEM FORMULATION

Consider a scenario where $N$ clients participate in a Federated Learning (FL) training process. Each client $i$ has a local dataset $D_i$ following distribution $P_i$ and a loss function $F_i(D_i, w_i)$, where $w_i$ represents the weight of its model $M_i$. In standard FL, all clients and the server share the same model architecture, i.e., $M_1 = M_2 = \cdots = M_N = M$. The objective of FL is to solve the following optimization problem:

$$\min_{w \in \mathbb{R}^d} G(F_1(D_1, w), F_2(D_2, w), \ldots, F_N(D_N, w)), \tag{1}$$

where $G(\cdot)$ is an aggregation function, typically an arithmetic mean or a weighted average based on dataset sizes $|D_i|$. The global model $w$ at communication round $t$ is updated as:

$$w^{t+1} = G(w_1^t, w_2^t, \ldots, w_N^t) = w^t - G(g_1^t, g_2^t, \ldots, g_N^t), \tag{2}$$

where $g_i^t = w^t - w_i^t$ represents the model weight difference after locally training for $K$ rounds. To reduce communication overhead, a compressor $C$ is applied:

$$w^{t+1} = w^t - G(C(g_1^t), C(g_2^t), \ldots, C(g_N^t)), \tag{3}$$

The goal of compression is to minimize the error:

$$C^* = \arg\min \|C(g_i^t) - g_i^t\|^2, \quad \text{s.t.} \quad \|C(g_i^t)\|_0 \leq B, \tag{4}$$

where $B$ is the communication budget. Let $\epsilon_i^t = \|C(g_i^t) - g_i^t\|$ denote the compression error. Error feedback is used to optimize this error by adding it to $g_i^{t+1}$:

$$w^{t+1} = w^t - G(C(g_1^t + \epsilon_1^t), C(g_2^t + \epsilon_2^t), \ldots, C(g_N^t + \epsilon_N^t)), \tag{5}$$

with compression error updated as:

$$\epsilon_i^{t+1} = g_i^t + \epsilon_i^t - C(g_i^t + \epsilon_i^t). \tag{6}$$

In 3SFC, each client first trains its local model using its dataset. Local updates are computed as the difference between the global model and the locally trained model weights. The encoder compresses the updates into a synthetic dataset $D_t^{\text{syn},i}$ within the communication budget. The server reconstructs and aggregates the gradients before updating the global model.

The encoder's objective is to minimize the following loss:

$$\min_{D_t^{\text{syn},i}, s_i^t} \left\| s_i^t \nabla_{w_t} F_i(D_t^{\text{syn},i}, w_t) - g_i^t - \epsilon_i^t \right\|^2 + \lambda \|D_t^{\text{syn},i}\|^2, \tag{7}$$

subject to:

$$\|D_t^{\text{syn},i}\|_0 + 1 \leq B. \tag{8}$$

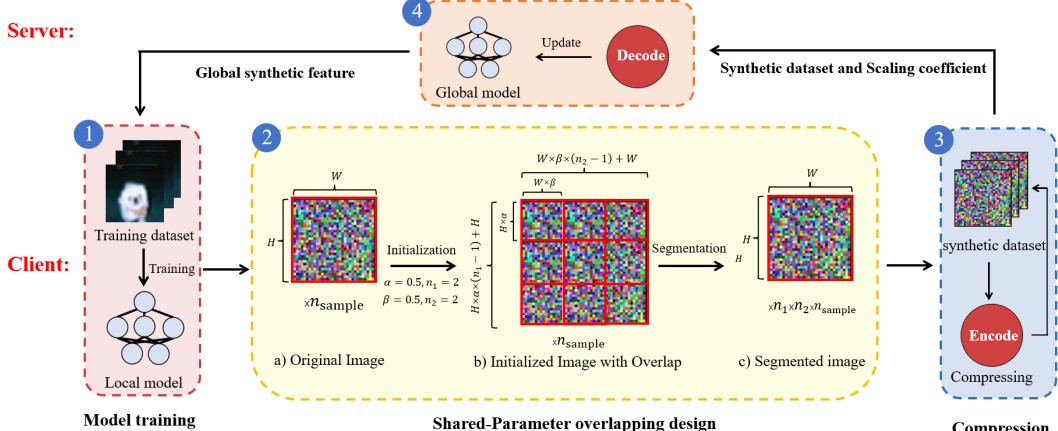

Figure 2: The general architecture of OFS involving a server and client.On the server side, the system manages global synthetic features and the global model, and updates the global model using synthetic datasets and corresponding scaling coefficients received from the clients.On the client side, the system processes local training datasets and compresses accumulated gradients using a shared-parameter overlapping design.

The scaling coefficient $s_i^t$ is computed as:

$$s_i^t = \frac{\|g_i^t + \epsilon_i^t\|}{\|\nabla_{w_t} F_i(D_t^{\mathrm{syn},i}, w_t)\|} \cos(\theta), \tag{9}$$

where $\theta$ is the angle between $g_i^t + \epsilon_i^t$ and $\nabla_{w_t} F_i(D_t^{\mathrm{syn},i}, w_t)$. Once $s_i^t$ is determined, the compression error $\epsilon_i^t$ is updated, and $D_t^{\mathrm{syn},i}$ and $s_i^t$ are uploaded to the server.

The server reconstructs the gradients for local model updates:

$$g_i^t + \epsilon_i^t = s_i^t \nabla_{w_t} F_j(D_t^{\mathrm{syn},i}, w_t). \tag{10}$$

The reconstructed gradients are used to update the global model.

## 4 OUR APPROACH

Building upon the traditional 3SFC method, we introduce a shared-parameter overlapping design, as illustrated in Figures 2, to optimize communication efficiency and gradient compression strategies.

During each training epoch, the $i$-th client first trains its local model using its local dataset. After local training steps, the accumulated gradients are computed as the difference between the local model weights and the global model weights. Then, the $i$-th client initializes a synthetic dataset $D_{\mathrm{syn},i}^t$ and decomposes it according to the overlapping strategy, yielding $D_{\mathrm{syn},i}^t{}'$. Next, the client utilizes an encoder to compress the data into $D_{\mathrm{syn},i}^t{}'$, ensuring it meets the communication budget. Once compression is completed, the client reassembles it into $D_{\mathrm{syn},i}^t$ based on the overlapping scheme and uploads it to the server.

Upon receiving the compressed data, the server first decomposes the synthetic dataset $D_{\mathrm{syn},i}^t$ using the same overlapping strategy to obtain $D_{\mathrm{syn},i}^t{}'$. It then decodes these synthetic data to faithfully reconstruct the accumulated gradients, performs gradient aggregation across all clients, and subsequently updates the global model accordingly.

### 4.1 SHARED-PARAMETER OVERLAPPING DESIGN

Throughout the communication process, we employ a shared-parameter overlapping design to optimize the original method. The overall architecture is illustrated in Figure 2. Specifically, we introduce the following parameters:

- $\alpha$: Offset value in the $H$ (height) direction.

- $n_1$: Number of overlapping images in the $H$ direction.

- $\beta$: Offset value in the $W$ (width) direction.

- $n_2$: Number of overlapping images in the $W$ direction.

- $n_{sample}$: Number of images in $D_{\text{syn},i}^t$.

The relationship between these parameters and the communication cost is given by:

$$B = C \times \big(H \times \alpha \times (n_1 - 1) + H\big) \times \big(W \times \beta \times (n_2 - 1) + W\big) \times n_{\text{sample}}. \tag{11}$$

The synthetic dataset $D_{\text{syn},i}^t$ in client $i$ and its decomposed

$$D_{\text{syn},i}^t = \{P_j \mid P_j \in \mathbb{R}^{C \times (H \times \alpha \times (n_1 - 1) + H) \times (W \times \beta \times (n_2 - 1))}, j = 1, 2, \ldots, n_{\text{sample}}\}. \tag{12}$$

$$D_{\text{syn},i}^t{}' = \{P_j \mid P_j \in \mathbb{R}^{C \times H \times W}, \quad j = 1, 2, \ldots, n_1 \times n_2 \times n_{\text{sample}}\}. \tag{13}$$

That is, by adjusting the values of $\alpha, \beta, n_1, n_2$, we can effectively achieve image overlapping and parameter sharing while meeting the communication cost $B$. The detailed process is as follows:

First, we initialize the parameters $\alpha, \beta, n_1, n_2$, and $n_{\text{sample}}$. Then, on the $i$-th client, a synthetic dataset $D_{\text{syn},i}^t$ is generated that satisfies the communication cost $B$.

This dataset consists of $n_{\text{sample}}$ images, each of size $C \times (H \times \alpha \times (n_1 - 1) + H) \times (W \times \beta \times (n_2 - 1) + W)$. Subsequently, according to the defined decomposition strategy, the features are divided into $n_{\text{sample}} \times n_1 \times n_2$ image patches of size $C \times H \times W$. For instance, along the height dimension $H$, adjacent patches are overlapped by a ratio of $\alpha$, repeated for $n_1$ patches in that direction. As illustrated in Figure 2.2, these overlapping patches collectively constitute the synthesized dataset $D_{\text{syn},i}^t{}'$.

During the parameter update process, when the gradient of each image is updated, the shared parameters in the overlapping regions are also synchronously updated. After the update is completed, the dataset $D_{\text{syn},i}^t{}'$ is reassembled into $D_{\text{syn},i}^t$ following the previous overlapping strategy and then transmitted to the server. The server decomposes $D_{\text{syn},i}^t$ in the same manner, performs gradient aggregation, and updates the global model.

Through this approach, redundant parameters are effectively reduced. The introduced parameters $\alpha, \beta, n_1, n_2$ transform the originally discrete compression operation into a quantifiable process, thereby improving compression efficiency, reducing communication overhead, and further optimizing the gradient transmission process between clients.

---

**Algorithm 1** Overlapped Feature Synthesis (One Epoch)

---

**Input:** global model $w_t$, local dataset $D_i$, learning rate $\eta_i$, accumulated gradient $\epsilon_i^t$, regularization parameter $\lambda$
**Parameters:** communication budget $B$, local iterations $K$, 3SFC iterations $S$, clients $N$, aggregation $G$, overlapping design $(\alpha, n_1; \beta, n_2)$, sample size $n_{\text{sample}}$
**Output:** global model $w_{t+1}$

1: **for** each client $i$ from 1 to $N$ in parallel **do**
2:   Initialize $D_{\text{syn},i}^t$
3:   **for** each local iteration $e$ from 1 to $K$ **do**
4:    Update local model $w_i^t$
5:   **end for**
6:   Compute local gradient $g_i^t = w_i^t - w_t$
7:   **for** each 3SFC iteration $s$ from 1 to $S$ **do**
8:    Encode synthetic data $D_{\text{syn},i}^t{}'$
9:   **end for**
10:   Compute step size $s_i^t$
11:   Update accumulated gradient $\epsilon_i^{t+1}$
12:   Merge $D_{\text{syn},i}^t{}'$ according to overlapping design
13:   Return $D_{\text{syn},i}^t, s_i^t, \epsilon_i^{t+1}$
14: **end for**

**Server:**

1: **for** each client $i$ from 1 to $N$ **do**
2:   Receive $D_{\text{syn},i}^t, s_i^t$
3:   Segment and Decode $D_{\text{syn},i}^t{}'$
4: **end for**
5: Aggregate updates $w_{t+1} = w_t - G(\ldots)$
6: Return $w_{t+1}$

---

## 4.2 ENCODER AND DECODER WITH OVERLAPPING DESIGN

Our encoder and decoder architecture is based on the state-of-the-art 3SFC method in data distillation, with modifications. The encoder's primary task is to compress the gradient into a synthetic dataset $D_{\text{syn},i}^t$ and a scaling factor $s_i^t$; the decoder's main task is to reconstruct the gradients by decoding the compressed data. For brevity, we do not reiterate those details here. Considering that we employ an overlapping design, the construction of the synthetic dataset $D_{\text{syn},i}^t$ has changed.

Before encoding, we first initialize the synthetic dataset $D_{\text{syn},i}^t$ based on the overlapping strategy described in Section 4.1, and then decompose it into $D_{\text{syn},i}^t{}'$ following the decomposition strategy.

According to the overlapping design, each $j$-th image $D_{\text{syn},i,j}^t{}'$ in the decomposed synthetic dataset $D_{\text{syn},i}^t{}'$ consists of a global part and a personalized part from the $x$-th image $D_{\text{syn},i,x}^t$ in the original synthetic dataset before decomposition, expressed as:

$$D_{\text{syn},i,j}^t{}' = D_{\text{globe},i,x}^t + D_{\text{personalize},i,x}^t \tag{14}$$

During subsequent processing, we consistently use the decomposed synthetic dataset $D_{\text{syn},i}^t{}'$. We update it according to the 3SFC approach to obtain the updated $D_{\text{syn},i}^t{}'$ and scaling coefficient $s_i^t$, which contains the accumulated gradient along with the offset coefficient. Finally, a merging operation is performed to reassemble $D_{\text{syn},i}^t{}'$ back into $D_{\text{syn},i}^t$, preserving the original dataset structure.

Upon receiving the synthetic dataset $D_{\text{syn},i}^t$ and scaling factor $s_i^t$ from clients, the server similarly decomposes $D_{\text{syn},i}^t$ into $D_{\text{syn},i}^t{}'$ according to the same strategy. The server then reconstructs the gradients for local model updating using the 3SFC approach.

The proposed **overlapping design** effectively integrates **gradient compression** and **parameter sharing**, thereby improving communication efficiency and reducing transmission overhead. Although introduced within the 3SFC framework, this design is not limited to 3SFC and can be extended to other federated learning paradigms to further enhance their communication performance.

## 5 EXPERIMENT

**Datasets:** We use five standard benchmark datasets: MNIST Deng (2012), FMNIST Xiao et al. (2017), EMNIST Cohen et al. (2017), CIFAR-10, and CIFAR-100 Krizhevsky et al. (2009). To simulate non-independent and identically distributed (non-i.i.d.) characteristics, all datasets are partitioned using a Dirichlet distribution Zhao et al. (2018); Wang et al. (2020b); Li et al. (2022); Yurochkin et al. (2019), ensuring heterogeneity in both data volume and class distribution.

**Models:** To cover tasks from simple to complex, this experiment selects five models, including a Multilayer Perceptron (MLP), MnistNet, ConvNet Caldas et al. (2018), ResNetHe et al. (2016), and RegNetRadosavovic et al. (2020). Specifically, MnistNet consists of two convolutional layers and two fully connected layers; ConvNet includes four convolutional layers and one fully connected layer. For ResNet and RegNet, all batch normalizationIoffe & Szegedy (2015) and dropout layersSrivastava et al. (2014) are removed, as these layers' parameters are not trainableWang et al. (2023). This simplification strategy has been applied in previous researchZhou et al. (2021); Sattler et al. (2019) and aims to reduce model complexity and accelerate training.

**Competetors:** We compare the proposed method with five other methods: FedAvgMcMahan et al. (2017), DGCLin et al. (2017), signSGD with EFBernstein et al. (2018), z-signTang et al. (2024), and 3SFCZhou et al. (2023). FedAvg McMahan et al. (2017) is a baseline without compression. DGC is regarded as the current state-of-the-art sparse method. signSGD is a typical gradient quantization method, while z-sign represents one of the most advanced and effective sign-based compression algorithms currently available, offering improved convergence and communication efficiency. 3SFC is the leading approach in data distillation-based compression. All experiments are conducted under 10, 20, and 40 client settings.

| Methods | MNIST | EMNIST | FMNIST | | Cifar10 | | | Cifar100 | |
|---|---|---|---|---|---|---|---|---|---|
| | MLP | MLP | MLP | Mnistnet | ConvNet | ResNet | RegNet | ResNet | RegNet |
| | | | | | 10 Clients | | | | |
| FedAvg | 0.8915(1.0×) | 0.5757(1.0×) | 0.8003(1.0×) | 0.8498(1.0×) | 0.6117(1.0×) | 0.4177(1.0×) | 0.4048(1.0×) | 0.0900(1.0×) | 0.0923(1.0×) |
| signSGD | 0.8407(32.0×) | 0.4720(32.0×) | 0.7178(32.0×) | 0.7994(32.0×) | 0.6014(32.0×) | 0.3132(32.0×) | 0.2430(32.0×) | 0.0152(32.0×) | 0.0309(32.0×) |
| z-sign | 0.8338(32.0×) | 0.4680(32.0×) | 0.7180(32.0×) | 0.7996(32.0×) | 0.6027(**32.0×**) | 0.3124(32.0×) | 0.2432(32.0×) | 0.0152(32.0×) | 0.0308(32.0×) |
| DGC | 0.8590(125.0×) | 0.4903(125.0×) | 0.7467(125.0×) | 0.8176(666.7×) | 0.6074(5.2×) | 0.2407(1785.7×) | 0.3004(378.8×) | 0.0129(1785.7×) | 0.0418(378.8×) |
| 3SFC | 0.8798(125.0×) | 0.5314(125.0×) | 0.7773(125.0×) | 0.8201(666.7×) | 0.5990(5.2×) | **0.3312**(1785.7×) | 0.3850(378.8×) | 0.0438(1785.7×) | 0.0737(378.8×) |
| **OFS** | **0.8827**(125.0×) | **0.5535**(125.0×) | **0.7888**(125.0×) | **0.8253**(666.7×) | **0.6100**(5.2×) | 0.3205(**1785.7×**) | **0.3920**(378.8×) | **0.0498**(1785.7×) | **0.0781**(378.8×) |
| | | | | | 20 Clients | | | | |
| FedAvg | 0.8911(1.0×) | 0.5762(1.0×) | 0.7958(1.0×) | 0.8446(1.0×) | 0.6139(1.0×) | 0.4081(1.0×) | 0.4229(1.0×) | 0.1113(1.0×) | 0.1037(1.0×) |
| signSGD | 0.8433(32.0×) | 0.4857(32.0×) | 0.7022(32.0×) | 0.7919(32.0×) | 0.5975(32.0×) | 0.3134(32.0×) | 0.2608(32.0×) | 0.0223(32.0×) | 0.0438(32.0×) |
| z-sign | 0.8384(32.0×) | 0.4802(32.0×) | 0.7022(32.0×) | 0.7915(32.0×) | 0.5984(**32.0×**) | 0.3107(32.0×) | 0.2609(32.0×) | 0.0178(32.0×) | 0.0439(32.0×) |
| DGC | 0.8670(125.0×) | 0.4976(125.0×) | 0.7524(125.0×) | 0.8053(666.7×) | **0.6130**(5.2×) | 0.2540(1785.7×) | 0.3258(378.8×) | 0.0190(1785.7×) | 0.0444(378.8×) |
| 3SFC | 0.8827(125.0×) | 0.5370(125.0×) | 0.7803(125.0×) | 0.8124(666.7×) | 0.6021(5.2×) | 0.2916(1785.7×) | 0.3986(378.8×) | 0.0585(1785.7×) | 0.0784(378.8×) |
| **OFS** | **0.8848**(125.0×) | **0.5574**(125.0×) | 0.7873(125.0×) | **0.8208**(666.7×) | 0.6050(5.2×) | **0.3136**(1785.7×) | **0.4043**(378.8×) | **0.0649**(1785.7×) | **0.0833**(378.8×) |
| | | | | | 40 Clients | | | | |
| FedAvg | 0.8892(1.0×) | 0.5782(1.0×) | 0.7933(1.0×) | 0.8453(1.0×) | 0.6089(1.0×) | 0.4099(1.0×) | 0.4188(1.0×) | 0.1041(1.0×) | 0.0862(1.0×) |
| signSGD | 0.8487(32.0×) | 0.4946(32.0×) | 0.7197(32.0×) | 0.7949(32.0×) | 0.5971(32.0×) | 0.3124(32.0×) | 0.2631(32.0×) | 0.0135(32.0×) | 0.0355(32.0×) |
| z-sign | 0.8418(32.0×) | 0.4913(32.0×) | 0.7196(32.0×) | 0.7939(32.0×) | 0.5974(**32.0×**) | 0.3118(32.0×) | 0.2632(32.0×) | 0.0135(32.0×) | 0.0353(32.0×) |
| DGC | 0.8633(125.0×) | 0.5004(125.0×) | 0.7458(125.0×) | 0.8221(666.7×) | **0.6080**(5.2×) | 0.2630(1785.7×) | 0.3175(378.8×) | 0.0149(1785.7×) | 0.0500(378.8×) |
| 3SFC | 0.8774(125.0×) | 0.5426(125.0×) | 0.7726(125.0×) | 0.8192(666.7×) | 0.5974(5.2×) | **0.3261**(1785.7×) | 0.4011(378.8×) | 0.0712(1785.7×) | 0.0612(378.8×) |
| **OFS** | **0.8808**(125.0×) | **0.5627**(125.0×) | **0.7824**(125.0×) | **0.8252**(666.7×) | 0.6010(5.2×) | 0.3219(1785.7×) | **0.4050**(378.8×) | **0.0783**(1785.7×) | **0.0647**(378.8×) |

Table 1: Comparison of test accuracy and compression ratio. Note that OFS, 3SFC, and DGC have much higher compression ratios compared to signSGD and z-sign due to the limitations of quantization-based methods and the high compression efficiency of OFS, 3SFC, and DGC. Under the same compression ratio, OFS achieves better performance than 3SFC. A dedicated comparison between OFS and 3SFC is presented in Section 6.2 to further demonstrate the superiority of OFS.

# 6 ANALYSIS

## 6.1 PERFORMANCE COMPARISONS

We begin by comparing the final test accuracy of Overlapped Feature Synthesis (OFS) with several competing methods after 100 training epochs. In these experiments, the learning rate was set to 0.01. The batch sizes were configured as 512, 256, and 128 for settings with 10, 20, and 40 clients, respectively. The number of local iterations K was fixed at 5, and the regularization coefficient $\lambda$ was set to 0 (i.e., no regularization was applied).

To ensure fairness, the compression rates of DGC and 3SFC were matched to that of OFS, while quantization-based baselines (e.g., signSGD and z-sign) were fixed at a rate of 1/32 and compared with OFS under identical settings.

Table 1 presents the overall accuracy results. The findings demonstrate that, under equivalent compression constraints, OFS consistently outperforms DGC and 3SFC in terms of final test accuracy. This suggests that OFS enables more efficient training and faster convergence when communication resources are limited. On the other hand, OFS consistently delivers better model performance than signSGD and z-sign, even though these methods involve substantially greater communication over-

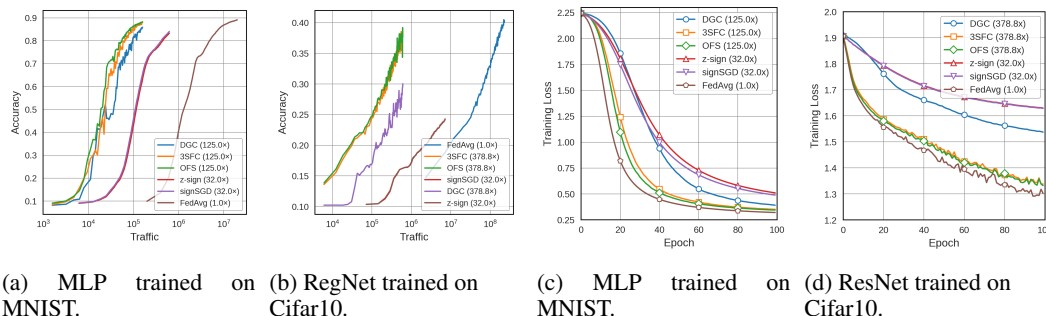

(a) MLP trained on MNIST.  (b) RegNet trained on Cifar10.  (c) MLP trained on MNIST.  (d) ResNet trained on Cifar10.

Figure 3: Test accuracy and training loss comparisons after 100 epochs of training. Compared to other methods, OFS owns the fastest convergence rate with respect to the amount of traffic communicated, with the highest compression ratio.

| Methods | MNIST | EMNIST | FMNIST | | Cifar10 | | | Cifar100 | | Statistics | |
|---|---|---|---|---|---|---|---|---|---|---|---|
| | MLP | MLP | MLP | Mnistnet | ConvNet | ResNet | RegNet | ResNet | RegNet | Avg | Std |
| | | | | | 10 Clients | | | | | | |
| 3SFC(2×B) | 0.8817 | 0.5408 | 0.8054 | 0.8401 | 0.5901 | 0.2955 | 0.4248 | 0.0473 | 0.0790 | 0.5005 | 0.3150 |
| OFS(2×B) | **0.8959** | **0.5698** | **0.8184** | **0.8429** | **0.5922** | **0.2998** | **0.4434** | **0.0638** | **0.0960** | 0.5136 | 0.3131 |
| OFS(1.75×B) | 0.8896 | 0.5576 | 0.8159 | 0.8376 | 0.5867 | 0.2949 | 0.4389 | 0.0618 | 0.0872 | 0.5078 | 0.3131 |
| OFS(1.5×B) | 0.8801 | 0.5447 | 0.8147 | 0.8321 | 0.5789 | 0.2986 | 0.4303 | 0.0520 | 0.0835 | 0.5016 | 0.3126 |
| | | | | | 20 Clients | | | | | | |
| 3SFC(2×B) | 0.8782 | 0.4635 | 0.7757 | 0.8130 | 0.5549 | **0.2992** | 0.3970 | 0.0329 | 0.0570 | 0.4746 | 0.3124 |
| OFS(2×B) | **0.8884** | **0.5286** | **0.7956** | **0.8175** | **0.5603** | 0.2950 | **0.4159** | **0.0570** | **0.0795** | 0.4931 | 0.3093 |
| OFS(1.75×B) | 0.8830 | 0.5055 | 0.7900 | 0.8143 | 0.5514 | 0.2949 | 0.4109 | 0.0527 | 0.0775 | 0.4867 | 0.3081 |
| OFS(1.5×B) | 0.8756 | 0.4725 | 0.7799 | 0.8083 | 0.5472 | 0.2940 | 0.3988 | 0.0472 | 0.0621 | 0.4762 | 0.3086 |
| | | | | | 40 Clients | | | | | | |
| 3SFC(2×B) | 0.8427 | 0.3794 | 0.6903 | 0.7928 | 0.5179 | 0.2750 | 0.3484 | 0.0362 | 0.0478 | 0.4367 | 0.2982 |
| OFS(2×B) | **0.8582** | **0.4779** | **0.7488** | **0.8046** | **0.5246** | 0.2724 | **0.3830** | **0.0476** | **0.0570** | 0.4638 | 0.3039 |
| OFS(1.75×B) | 0.8465 | 0.4430 | 0.7351 | 0.8011 | 0.5169 | 0.2724 | 0.3740 | 0.0492 | 0.0533 | 0.4546 | 0.3003 |
| OFS(1.5×B) | 0.8294 | 0.4246 | 0.7043 | 0.7970 | 0.5043 | **0.2756** | 0.3616 | 0.0268 | 0.0526 | 0.4418 | 0.2973 |

Table 2: Test accuracy and compression ratio comparisons of OFS and 3SFC with different communication budgets (transposed version).

head. Additionally, Figure 3 further illustrates the advantages of OFS by plotting the trends of test accuracy and training loss over time.

## 6.2 FURTHER COMPARISONS BETWEEN OFS AND 3SFC

To comprehensively assess the performance differences between OFS and 3SFC, we conducted a multi-dimensional comparison under consistent experimental settings — a learning rate of 0.01, batch size of 64, local iteration number $K = 5$, and no regularization. By varying the communication cost of OFS, we systematically analyzed its performance across scenarios with 10, 20, and 40 clients, as summarized in Table 2. The results indicate that, under equivalent compression rates (2×B), OFS exhibits significant advantages over 3SFC. For instance, in the CIFAR-100 with ResNet task, OFS-2×2-0.25 (0.1201) outperforms 3SFC-4×1 (0.1055) by 13.8%. When the communication cost of OFS is further reduced to 1.75×B (i.e., a 1.14-fold increase in compression rate), it still outperforms 3SFC in over 80% of the experiments. Furthermore, we explored the maximum compression rate that OFS can tolerate without compromising model accuracy. The results show that even with a 1.33-fold increase in the compression rate (communication cost reduced to 1.5×B), OFS maintains comparable performance. These findings confirm the superiority of OFS in balancing communication efficiency and model accuracy, offering an effective solution for bandwidth-constrained distributed learning. The results clearly show that OFS achieves superior performance even with significantly lower communication budget, highlighting its effectiveness in communication-efficient federated learning.

## 6.3 IMPACT OF OFFSET COEFFICIENT ON OFS PERFORMANCE

To explore the influence of the offset coefficient $\alpha$ on the performance of Overlapped Feature Synthesis (OFS), we fixed the communication cost at 1.5×B and 3×B, and evaluated three offset coefficient settings: $\alpha = 0.1, 0.25, 0.5$, corresponding to global-to-private parameter ratios of 9:1, 3:1, and 1:1, respectively. The experimental results are summarized in Table 3. The results indicate that, in most cases, a higher proportion of global parameters (i.e., smaller $\alpha$) leads to significantly better model performance. For instance, in the CIFAR-100-RegNet task, setting $\alpha = 0.1$ yield an accuracy of 0.1244, which is 24.8% higher than the 0.0997 accuracy achieved at $\alpha = 0.5$. This suggest that complex tasks rely more heavily global features. However, certain tasks achieve peak performance when $\alpha = 0.25$. For instance, in the CIFAR-10-ConvNet task, the accuracy at $\alpha = 0.5$ (0.6055) marginally exceeds that at $\alpha = 0.25$ (0.6053). Additionally, we observed that when $\alpha \leq 0.25$, the accuracy fluctuation across all tasks remained within 2%. Based on these findings, we recommend using $\alpha < 0.25$ by default——particularly for complex models——while $\alpha = 0.25$ remains a viable choice for lightweight tasks.

| Methods | MNIST | EMNIST | FMNIST | | Cifar10 | | | Cifar100 | | Statistics | |
|---|---|---|---|---|---|---|---|---|---|---|---|
| | MLP | MLP | MLP | Mnistnet | ConvNet | ResNet | RegNet | ResNet | RegNet | Avg | Std |
| | | | | | 1.5×B | | | | | | |
| OFS($\alpha$=0.1) | **0.8886** | **0.5609** | **0.8176** | 0.8269 | **0.5905** | 0.2928 | **0.4512** | **0.0631** | **0.1102** | 0.5113 | 0.3077 |
| OFS($\alpha$=0.25) | 0.8801 | 0.5447 | 0.8147 | **0.8321** | 0.5789 | **0.2986** | 0.4303 | 0.0520 | 0.0835 | 0.5016 | 0.3126 |
| OFS($\alpha$=0.5) | 0.8756 | 0.5335 | 0.8000 | 0.8288 | 0.5787 | 0.2934 | 0.4145 | 0.0460 | 0.0761 | 0.4941 | 0.3126 |
| | | | | | 2×1.5×B | | | | | | |
| OFS($\alpha$=0.1) | **0.9046** | **0.6009** | 0.8266 | 0.8487 | **0.6090** | **0.3046** | **0.4795** | **0.0759** | **0.1244** | 0.5305 | 0.3093 |
| OFS($\alpha$=0.25) | 0.9018 | 0.5863 | **0.8271** | **0.8547** | 0.6053 | 0.2993 | 0.4663 | 0.0697 | 0.1061 | 0.5241 | 0.3141 |
| OFS($\alpha$=0.5) | 0.8987 | 0.5646 | 0.8228 | 0.8530 | 0.6055 | 0.3020 | 0.4674 | 0.0572 | 0.0997 | 0.5190 | 0.3155 |

Table 3: Test accuracy of OFS with different offset coefficient. As the offset coefficient decreases, OFS achieves higher test accuracy.

| Methods | MNIST | EMNIST | FMNIST | | Cifar10 | | | Cifar100 | | Statistics | |
|---|---|---|---|---|---|---|---|---|---|---|---|
| | MLP | MLP | MLP | Mnistnet | ConvNet | ResNet | RegNet | ResNet | RegNet | Avg | Std |
| | | | | | K = 1 | | | | | | |
| OFS (1.5×B) | 0.7937 | 0.3855 | 0.6874 | 0.7860 | 0.4976 | 0.2617 | 0.3406 | 0.0400 | 0.0495 | 0.4269 | 0.2883 |
| OFS (1.75×B) | 0.8225 | 0.4205 | 0.7082 | 0.7932 | 0.5106 | 0.2549 | 0.3571 | 0.0415 | 0.0536 | 0.4402 | 0.2953 |
| OFS (2×B) | 0.8429 | 0.4394 | 0.7248 | 0.7958 | 0.5236 | 0.2605 | 0.3544 | 0.0448 | 0.0550 | 0.4490 | 0.3001 |
| OFS (3×B) | 0.8544 | 0.4707 | 0.7593 | 0.8023 | 0.5462 | 0.2672 | 0.3646 | 0.0445 | 0.0622 | 0.4635 | 0.3056 |
| OFS (3.5×B) | 0.8648 | 0.4954 | 0.7679 | 0.8042 | 0.5525 | 0.2622 | 0.3723 | 0.0453 | 0.0664 | 0.4701 | 0.3082 |
| OFS (4×B) | 0.8712 | 0.5111 | 0.7731 | 0.8074 | 0.5591 | 0.2584 | 0.3739 | 0.0455 | 0.0697 | 0.4744 | 0.3104 |
| | | | | | K = 10 | | | | | | |
| OFS (1.5×B) | 0.8951 | 0.5825 | 0.8288 | 0.8571 | 0.5744 | 0.3029 | 0.4631 | 0.0680 | 0.0976 | 0.5188 | 0.3142 |
| OFS (1.75×B) | 0.9010 | 0.5992 | 0.8269 | 0.8622 | 0.5835 | 0.3128 | 0.4720 | 0.0804 | 0.1093 | 0.5275 | 0.3109 |
| OFS (2×B) | 0.9043 | 0.6153 | 0.8315 | 0.8671 | 0.5801 | 0.3080 | 0.4750 | 0.0874 | 0.1144 | 0.5314 | 0.3113 |
| OFS (3×B) | 0.9095 | 0.6230 | 0.8377 | 0.8714 | 0.5891 | 0.3318 | 0.4949 | 0.0932 | 0.1252 | 0.5418 | 0.3085 |
| OFS (3.5×B) | 0.9137 | 0.6432 | 0.8398 | 0.8730 | 0.5988 | 0.3124 | 0.5002 | 0.1153 | 0.1316 | 0.5475 | 0.3071 |
| OFS (4×B) | 0.9146 | 0.6505 | 0.8431 | 0.8766 | 0.6008 | 0.3253 | 0.5057 | 0.1198 | 0.1449 | 0.5535 | 0.3041 |

Table 4: The ablation study with different parameters of OFS(i.e.,communication budgets B, local iteration K). The configuration for the Base is clients=10. From the table, increasing B or K can both further boost the convergence rate of the training.

## 6.4 ABLATION STUDY

Table 4 presents the results of an ablation study on OFS, focusing on the communication budget $B$ and the local iteration number $K$. The table shows that increasing the communication budget $B$ significantly improves the model's test accuracy, as it allows more data to be transmitted per communication round. Conversely, reducing the local iteration number $K$ from 5 to 1 leads to a significant decrease in test accuracy, as fewer model optimization iterations are performed. when $K$ s increased to 10, the test accuracy improves. Therefore, increasing the communication budget $B$ and adjusting the local iteration number $K$ are effective strategies for improving the convergence speed of Overlapped Feature Synthesis. However, in scenarios with a strictly limited communication budget, the convergence speed of Overlapped Feature Synthesis can still be improved by increasing the local iteration number $K$.

## 7 CONCLUSION

In this paper, we proposed Overlapped Feature Synthesis (OFS), a novel and efficient method for communication-efficient federated learning. By introducing a global feature sampler and an offset coefficient, OFS enables effective global feature sharing while maintaining personalized model capabilities. Our design allows for flexible control over the trade-off between communication efficiency and model performance through multiple sampling strategies. Comparisons of test accuracy and compression ratio show that OFS achieves significantly faster convergence rates with lower compression rates. An ablation study demonstrates the role of different parameters, and visualizations of compression efficiency further validates the effectiveness of OFS.

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

## A    APPENDIX

### A.1    THE USE OF LLMS

Large Language Models (LLMs), such as ChatGPT, were used in this work solely for language polishing and minor grammatical corrections. They were not involved in research ideation, methodological design, experimental implementation, analysis, or drawing of conclusions. All scientific contributions and decisions were made independently by the authors.

