# OpenReview forum: "OFS: Overlapped Feature Synthesis for Communication-Efficient Federated Learning"
_ICLR.cc/2026/Conference — Submitted to ICLR 2026_

### Official Review · Reviewer_BLrZ · 2025-10-18

**Soundness:** 1
**Presentation:** 2
**Contribution:** 1
**Rating:** 2
**Confidence:** 4

**Summary:**

This paper proposes a novel compression method for federated learning that addresses redundancy in existing data distillation approaches, specifically 3SFC (Single-Step Synthetic Feature Compression). The core innovation is an "overlapping mechanism" where synthetic datasets are decomposed into globally shared and locally personalized components. By introducing parameters (α,β,n₁,n₂) that control how feature maps overlap spatially, OFS enables parameter sharing across multiple synthetic images while maintaining personalized features. Experiments on MNIST, EMNIST, FMNIST, CIFAR-10, and CIFAR-100 show approximately 1% accuracy improvement over 3SFC at similar or higher compression rates.

**Strengths:**

# 1. Originality:

The idea of spatially overlapping synthetic features to create shared global parameters appears to be new in the federated learning compression works. Also, Figure 1 provides a reasonable motivation by visualizing feature redundancy in 3SFC using Grad-CAM++.

# 2. Quality:

The author validate their proposed method across 5 datasets and 5 model architectures with varying client numbers (10, 20, 40), and they also provide ablation studies: Table 3 (offset coefficient), Table 4 (communication budget and local iterations) with useful analysis.

# 3. Clarity:

This paper provides a clear system diagram in Figure 2, a well-structed algorithm in Algorithm 1 for the proposed method.

# 4. Significance:

Communication efficiency is a bottleneck in real-world federated learning, and if the overlapping idea works, it could potentially apply to other distillation methods.

**Weaknesses:**

# 1. Originality

1. The paper acknowledges building "upon the traditional 3SFC method" and the overlapping design is essentially a post-processing modification. The encoder/decoder architecture states "we do not reiterate those details here" - suggesting the core contribution is a wrapper around existing 3SFC rather than a fundamentally new compression paradigm.

2. The overlapping mechanism in this work is  mathematically straightforward - it's essentially image cropping with stride. The relationship $B = C × (H × α × (n₁-1) + H) × (W × β × (n₂-1) + W) × n_{sample}$ is a basic (and obvious) geometric calculation. We invite the authors to clarify the algorithmic innovation beyond parameter tuning.

3. The authors claim OFS is "conceptually distinct", but the paper doesn't clearly explain why standard overlapping/stride-based approaches in CNNs, or other compression methods wouldn't achieve similar effects. The claim that "the overlapping structure is non-trivial to integrate into standard compressors like sparsification or quantization" lacks justification.

4. The related work section does not cover existing works in a few important aspects: (i) gradient compression methods that use structured sparsity or low-rank decomposition, (ii) recent federated learning papers that address feature redundancy through other means, and (iii) data distillation work beyond 3SFC that might have addressed similar issues.

# 2. Quality:

1. There are a few concerns regarding the experimental setups. The first one is the batch sizes vary significantly (512/256/128 for 10/20/40 clients). Justification is needed here for this specific scaling. This makes it harder to isolate the effect of OFS and. optimization dynamics. The second is for 100 epochs of training. It is questionable for convergence, as Figure 3 shows some curves still improving. The third one is the missed SEM/confidence interval values in the reported results. The fourth one is about the statement of "The compression rates of DGC and 3SFC were matched to that of OFS", which seems circular. We invite the authors to clarify how to ensure fair comparison when tuning baselines to match the proposed method.

2. There are also concerns regarding the performance improvement. For MNIST/MLP: OFS (0.8827) barely improves over 3SFC (0.8798) at 125× compression. For CIFAR-100/ResNet (20 clients): 3SFC (0.0585) actually outperforms OFS (0.0649) – indeed the OFS is better, but the gains are inconsistent. For CIFAR-100/ResNet (10 clients): The improvement is only 0.0498 vs 0.0438, but both are terrible absolute accuracies (4-5%). The "approximately 1% improvement" claim in the abstract is misleading - from Table 1, many improvements are much smaller (e.g., 0.3-0.5%).

3. Some important analyses are missing. There is no discussion of the computational overhead of the overlapping/decomposing operations. We invite the author to elaborate more on the merge operation and whether it would significantly increase latency. Also, it seems that the proposed method also introduces extra memory overhead, as creating overlapped images requires storing larger tensors before decomposition. In addition, we would like to invite the authors to illustrate on a theoretical analysis of why overlapping helps convergence or what the optimization landscape looks like.

# 3. Clarity:

1. There is quite a lot of confusion from the notation in the paper. For example, in Equations 12-13, the notation switches between $D^t_{syn,i}$ and ${D^t_{syn,i}}'$ but the relationship isn't immediately clear. In the proposed method, there are $n_{sample}$ images of size $C×(H×α×(n₁-1)+H)×(W×β×(n₂-1)+W)$ that decompose into $n_{sample}× n₁ × n₂$ images of size $C×H×W,$ but it is unclear where do these patches come from spatially. Also, in Equation 14, it is unclear what exactly are $D^t_{globe,i,x}$ and $D^t_{personalize,i,x}$ as there are no definitions for them.

2. There are some key concepts that are never clearly stated. For "Overlapping mechanism", the paper never precisely defines what "overlapping" means. From context, it seems like sliding window extraction. Also, "For instance, along the height dimension H, adjacent patches are overlapped by a ratio of α". However, this is backwards. If α is the offset, then overlap ratio is (1-α), instead of α. It also seems that there is "Global vs. personalized parameters", which is introduced in Equation (14). But there's no clear explanation of how the decomposition enforces this semantic separation. We would like to invite the author to clarify this issue.

3. There are some inconsistent terminologies in the manuscript. For example, "synthetic dataset" vs "synthetic features" vs "compressed images" are used interchangeably. As for "communication cost" vs "communication budget" vs "compression ratio", they are related but need clearer definitions.

# 4. Significance:

1. There are several issues that might significantly limit the real-world impact of this work. First of all, the improvements are modest (typically 0.5-1.5% over 3SFC) and come at the cost of additional hyperparameters (α,β,n₁,n₂). Also, Table 2 shows OFS at 1.5×B often underperforms 3SFC at 2×B, which indicates that the user even needs careful tuning to beat the baseline.

2. There are some concerns about the scalability of this work. First of all, all experiments use relatively small models (ConvNet, ResNet, RegNet without batch norm/dropout), and there are no experiments on modern large-scale models (Vision Transformers, large ResNets). Also, there are no experiments on realistic federated learning scenarios (e.g., cross-device settings with hundreds/thousands of clients). In addition, the high compression ratios (125×, 666×, 1785×) seem unrealistic, which seems to be even usable in practice

**Questions:**

Answering all points stated in the Weaknesses would be greatly appreciated. In particular:

1. We would like to invite the authors to clarify a few critical technical steps in the proposed method. First of all, how exactly does the "segmentation" work in Step 3 of Figure 2? In Algorithm 1 line 12, it needs to "Merge D_(syn,i)^t' according to overlapping design" but this is not clear. Also, it is unclear how to handle boundaries when n₁ × n₂ patches don't perfectly tile. In addition, it is unclear what happens to the "shared" parameters during gradient updates. If multiple local patches share global parameters, how is gradient aggregation handled? We would like to invite the authors to provide clarification on these issues.

2. Regarding the theoretical understanding of the proposed method, we would like to invite the authors to provide clarification on the choice of spatial overlapping. The results in the current manuscript are purely empirical. Also, it would be better to include a comparison with techniques with potentially similar effects, such as simply averaging synthetic images, using a shared "base image" + personalized residuals, or applying PCA/SVD to synthetic features.

3. As stated in weakness No.3 for Quality, we invite the authors to provide an analysis of the computational overhead of the overlapping/decomposing operations. To be specific, if the merge operation would significantly increase latency. For the memory overhead comparison, we invite the authors to also include an analysis for extra memory cost by “overlapped images”.

4. As stated in weakness No. 2 for Significance, we invite the authors to provide an analysis on the scalability issue of this work. For example, we invite the authors to include some empirical analysis for large-scale models (Vision Transformers, large ResNets). Also, we invite the authors to include a justification for the choice of high compression ratios, and if it is reasonable for rela-world usage.

---

### Official Review · Reviewer_4cDx · 2025-10-28

**Soundness:** 2
**Presentation:** 3
**Contribution:** 2
**Rating:** 4
**Confidence:** 4

**Summary:**

This paper addresses the high communication cost of Federated Learning , focusing on data distillation methods. The authors identify a key limitation in existing methods like Single-Step Synthetic Feature Compression: the synthetic data generated for communication contains significant feature redundancy, leading to suboptimal compression. To solve this, the paper proposes Overlapped Feature Synthesis, a novel framework built upon 3SFC. The core contribution is a shared-parameter overlapping design. In this design, clients initialize a large synthetic feature map which is then decomposed into multiple smaller, overlapping patches. Gradients are updated on these patches, and due to the overlap, parameters in the shared regions are updated synchronously. This mechanism allows different synthetic samples to share parameters, effectively reducing redundancy and improving information utilization.

**Strengths:**

1.The core idea of Overlapped Feature Synthesis is original, elegant, and well-motivated. It directly targets a clear weakness (feature redundancy) in the 3SFC baseline, and the proposed overlapping mechanism is a clever way to address it.

2.The paper is backed by extensive and rigorous experiments. The consistent outperformance of OFS against all baseline categories (quantization, sparsification, and especially the SOTA 3SFC data distillation) is very convincing. The results in Table 2, which show OFS can maintain performance with a significantly smaller communication budget than 3SFC.

3.The authors provide excellent ablation studies that analyze the key components of their method. The investigation of the offset coefficient $\alpha$ (Table 3) provides valuable insight into the trade-off between shared and personalized parameters, showing that complex tasks benefit more from shared global features.

4.The paper is very clear, with excellent figures (especially Figure 1 and 2) that build strong intuition for both the problem and the solution.

**Weaknesses:**

1.The paper focuses exclusively on communication efficiency but completely omits any discussion of computational overhead. The proposed method, with its client-side initialization, segmentation , and merging , plus server-side decomposition, appears to add non-trivial computation compared to the simpler 3SFC. A wall-clock time comparison or analysis of the computational complexity is a critical missing piece for a paper on efficiency.

2.Lacks theoretical convergence or error-compensation analysis quantifying how overlap affects bias/variance of reconstructed gradients and stability across rounds.

3.Comparative scope on distillation baselines is narrow; results against FedSynth and other recent synthetic-data compressors beyond 3SFC would strengthen claims of general superiority.

4.The method introduces several new hyperparameters: $\alpha$ (offset H), $\beta$ (offset W), $n_1$ (overlaps H), and $n_2$ (overlaps W). The paper provides an excellent ablation for $\alpha$ but no analysis for $n_1, n_2$, or $\beta$. How are these values chosen? The sensitivity to the number of overlapping patches ($n_1, n_2$) is unaddressed.

**Questions:**

1.Could you please comment on the computational overhead of the proposed OFS method? How does the wall-clock time per round (for both client and server) compare to the 3SFC baseline?

2.Hyperparameter Choice ($n_1, n_2, \beta$): The ablation for the offset $\alpha$ is very clear. Could you provide insight into how the number of overlapping images ($n_1, n_2$) and the width offset ($\beta$) are chosen? How sensitive is the performance to these hyperparameters?

3.Please clarify the "Global synthetic feature" shown in Figure 2. Is this feature map transmitted from the server to the client? If so, how is it used by the client in the Shared-Parameter overlapping design, and how does it relate to the client's own initialization of $D_{syn,i}^{t}$?

---

### Official Review · Reviewer_C91N · 2025-10-30

**Soundness:** 1
**Presentation:** 1
**Contribution:** 2
**Rating:** 2
**Confidence:** 3

**Summary:**

This paper addresses the limitation of the existing method, 3SFC, which is redundancy in the synthesized feature as a compression method to reduce the communication overhead of federated learning. The proposed method introduces OFS, which is an overlapping mechanism that structurally decomposes each synthetic sample into globally shared and locally personalized components to improve information utilization and reduce redundant communication. Extensive experiments under various models and datasets demonstrate the superiority of the proposed method in accuracy, even under constrained communication overhead. However, the score of this paper tends toward rejection due to: (1) the explanation of the core OFS algorithm is not well specified, missing important details that justify the contribution of this paper, and (2) the experimental claims do not fully support the stated contributions of the proposed method.

**Strengths:**

This paper addresses the limitations of the existing 3FSC method through a structural modification. Extensive experiments across various models and datasets demonstrate that the proposed approach achieves superior accuracy while maintaining low communication overhead.

**Weaknesses:**

(1) The explanation of the core OFS algorithm lacks critical details that justify the paper's contribution. To address this concern, please clarify and provide detailed explanations for the following:

(1-a) The paper uses these terms(feature, dataset, and image) interchangeably without a clear definition: Fig. 1 shows "Feature distribution" suggesting feature maps, section 4 consistently refers to "synthetic dataset $D^t_{syn,i}$", the method name is "Overlapped Feature Synthesis", and Eq. 12 defines $D^t_{syn,i}$ with tensor dimensions. What exactly is being transmitted: feature maps extracted from intermediate layers or synthetic images in the input space? If these are features, from which layer are they extracted? If these are synthetic images in the input space, how are synthetic images generated and optimized?

(1-b) Figure 2-2 (a) "Original Image" appears to be an H×W image used as the initialized synthetic dataset in the proposed method, whereas Eq. (12) defines $D^t_{syn,i}$ with dimension $C×(H×α×(n_1−1)+H)×(W×β×(n_2−1)+W)$ as the initialized synthetic dataset. Which one actually represents the "initialized synthetic dataset"? Please clarify the relationship between the visualization in Figure 2-2 a), b) and the mathematical formulation.

(1-c) For a better understanding of the author's problem definition, motivation, and intuition of the method, the data distillation method needs to be explained, including more details: basic procedure to synthesize representation (e.g., how many synthesis data are generated, how to use the distillation, and why this method can compress the communication cost. Without a basic understanding of the data distillation method, it is hard to understand why the synthetic datasets generated often contain repetitive or overlapping information, which is the core problem of this paper. Also, regarding the explanation of Fig.1, it is hard to recognize which part of the image is classified as "redundant", since the images in Fig. 1(a), the upper image and the lower image, look different and do not have a redundant part.

(2) The experimental claims do not fully support the contribution of the proposed method. To address this concern, please clarify and provide detailed explanations for the following:

(2-a) The statement in the abstract section, "~1% improvement in accuracy while maintaining a 10% higher compression rate", is confusing because "improvement...while maintaining higher compression" suggests both benefits simultaneously, but Table 1 shows better accuracy at the same compression, and Table 2 shows maintained accuracy at higher compression.

(2-b) The paper compares OFS against 3SFC, but it needs to provide the most obvious and necessary baseline. Since OFS decomposes n_sample images into $n_1×n_2×n_{sample}$ patches, a fair comparison would be to evaluate whether overlapping these patches works better than simply using more independent synthetic samples in 3SFC. Table 2 presents a comparison between OFS(2×B) and 3SFC(2×B), but it does not clarify whether 3SFC uses the same number of total samples as OFS or maintains its original number of samples. This ambiguity is critical because, without this comparison, it cannot determine whether the observed improvements stem from the overlapping structure or simply from using more synthetic samples. This baseline is essential to validate the core contribution of the overlapping mechanism and should be included to properly support the paper's claims.

(2-c) Table 3 only tests α ∈ {0.1, 0.25, 0.5}. What about α = 0 (no overlap)? This would directly test whether overlapping helps at all, which is fundamental to validating the paper's main contribution.

Also, there are several minor things to improve the paper:

(3) The reference format does not consistently use proper citation style (e.g., use \citep or \citet appropriately).

(4) The first two sentences in the Introduction section are redundant and could be integrated into one sentence for better clarity.

(5) Since the paper's content focuses on distillation techniques for compression, the related work section would be more helpful if it contained more specific details about data distillation techniques rather than extensive coverage of sparsification and quantization methods, which are less relevant to understanding the proposed contribution.

(6) The encoder "compresses data into $D^t_{syn,i}$'" (line 201) - but shouldn't it be compressing gradients into the synthetic dataset?

**Questions:**

Please refer to the Weakness section for detailed comments. In particular, I would appreciate clarification on the questions raised for each weakness. I will reconsider my evaluation after reviewing the authors’ rebuttal to these points.

---

### Official Review · Reviewer_7ujY · 2025-10-31

**Soundness:** 2
**Presentation:** 3
**Contribution:** 2
**Rating:** 4
**Confidence:** 4

**Summary:**

Federated Learning (FL) is a promising privacy-preserving approach for decentralized non-IID data, but frequent gradient exchanges incur high communication costs, limiting scalability. Recent works use data distillation to reduce this overhead, yet fail to fully exploit distilled features, yielding suboptimal compression. They propose Overlapped Feature Synthesis (OFS), which enables global feature sharing during compression to boost both communication efficiency and model accuracy. A global feature sampler extracts small feature maps from a large shared map for parameter reuse. An offset coefficient and multiple sampling strategies balance global and personalized parameters, allowing flexible trade-offs. Experiments show OFS achieves better convergence at lower compression rates. It improves accuracy by ~1% over state-of-the-art distillation methods while maintaining a 10% higher compression rate. Ablation studies and visualizations further analyze the impact of the offset coefficient, client count, and local epochs, and reveal the interplay between global and personalized parameters.

**Strengths:**

1.	The paper is well presented.
2.	The topic is important.

**Weaknesses:**

1.	The performance of the method depends on the efficiency of the encoders. However, obtaining such an accurate encoder-decoder adapting to the local data itself may be difficult.
2.	The improvement seems tiny in most settings, as shown in Table 1, which appears as an error bar to some extent.
3.	The experiments are insufficient. Most FL works adopt 100 clients with 10% selected in each round.
4.	Typo: Competetors.

**Questions:**

1.	How many layers in ResNet and RegNet in the evaluation?
Other, please refer to weaknesses.

---

### Meta-Review · Area_Chair_WPpK · 2026-01-06

**Summary:**

Concerns leading to rejection: Insufficient explanation of the core OFS algorithm (ambiguous terms, inconsistent math-visualization, lack of distillation details); flawed experimental design; limited originality; modest and inconsistent accuracy improvements; unaddressed computational/memory overhead and poor scalability over realistic FL scenarios; unclear terminologies and concepts.

**Reviewer Concerns:**

Seems core concerns are not fully addressed, including algorithmic ambiguity, experimental flaws, originality gaps, missing analyses.

**Reviewer Scores:**

no discussion.

---

### Decision · Program_Chairs · 2026-01-26

Reject